# Surgery after Induction Targeted Therapy and Immunotherapy for Lung Cancer

**DOI:** 10.3390/cancers13112603

**Published:** 2021-05-26

**Authors:** Toon Allaeys, Lawek Berzenji, Paul E. Van Schil

**Affiliations:** Department of Thoracic and Vascular Surgery, Antwerp University Hospital, Drie Eikenstraat 655, B-2650 Edegem, Antwerp, Belgium; toon.allaeys@uza.be (T.A.); lawek.berzenji@uza.be (L.B.)

**Keywords:** non-small cell lung cancer, immunotherapy, targeted therapy, surgery

## Abstract

**Simple Summary:**

For patients with locally advanced non-small cell lung cancer (NSCLC) or positive N1 nodes, multimodality treatment is indicated. However, the optimal management of patients presenting with ipsilateral positive mediastinal nodes (N2 disease) has not been determined yet. Different treatment regimens consisting of chemotherapy, radiation therapy, and surgery have been proposed and implemented previously. In more recent years, immunotherapy and targeted therapies have been added as therapeutic options. The introduction of these newer modalities has raised questions on the role of surgery after targeted therapy or immunotherapy. Recent studies have shown that surgical resection after induction immunotherapy or targeted therapy is indeed feasible, but is associated with a higher risk of conversion and increased morbidity due to hilar inflammation. In this review, we summarize the latest data on outcomes of patients undergoing surgical resection after induction immunotherapy and/or targeted therapy. Treatment outcomes have to be carefully evaluated to determine the contribution of surgery in multimodality treatment regimens including immunotherapy and targeted therapies.

**Abstract:**

Multimodality therapy for locally advanced non-small cell lung cancer (NSCLC) is a complex and controversial issue, especially regarding optimal treatment regimens for patients with ipsilateral positive mediastinal nodes (N2 disease). Many trials investigating neoadjuvant immunotherapy and targeted therapy in this subpopulation have shown promising results, although concerns have risen regarding surgical feasibility. A thorough literature review was performed, analyzing all recent studies regarding surgical morbidity and mortality. Despite the fact that two major trials investigating this subject were terminated early, the overall consensus is that surgical management seems feasible. However, dissection of hilar vessels may be challenging due to hilar fibrosis. Further research is necessary to identify the role of surgery in these multimodality treatment regimens, and to define matters such as the optimal treatment regimen, the dosage of the different agents used, the interval between induction therapy and surgery, and the role of adjuvant therapy.

## 1. Introduction

Lung cancer is worldwide the most common malignant tumor and one of the leading causes of cancer-related death [1,2,3,4,5,6]. Non-small cell lung cancer (NSCLC) is the most common type of lung cancer, accounting for up to 85% of all lung cancers. Despite major improvements in imaging and treatment modalities, NSCLC is still associated with poor overall 5-year survival rates, mainly due to the fact that a complete surgical resection can only be obtained in a limited number of patients. Furthermore, many patients who got surgical treatment are at risk of developing a second primary tumor or recurrent disease due to micrometastases [2,4,6]. At time of diagnosis, the majority of patients have advanced stages of disease, with approximately 25% of all patients with NSCLC presenting with locally advanced disease and more than half with distant metastases. Outside screening programs, only 19% of all new NSCLC cases are diagnosed at an early-stage [3,6,7].

Treatment for NSCLC largely depends on the stage and extent of the disease. In early stage disease, lobectomy with hilar and mediastinal lymph node dissection is considered the gold standard. The value of adjuvant chemotherapy is not clear for lower stages. However, for resected stage IB disease with primary tumors >4 cm, adjuvant chemotherapy can be considered [8]. For advanced stages of NSCLC, surgery is often not recommended and chemoradiotherapy (CRT) is considered standard treatment. However, there is still a large grey zone when discussing optimal treatment for patients presenting with locally advanced disease. In the 8th edition of the tumor, node, and metastasis (TNM) staging system of the American Joint Committee on Cancer (AJCC), locally advanced disease is predominantly represented in stage III and is subdivided in stages IIIA, IIIB, and IIIC [9]. Some authors have suggested to subdivide stage III into resectable and non-resectable disease, although a generally accepted and precise definition of what constitutes resectable stage III is not available [3].

In contrast to early stage and metastatic NSCLC for which management guidelines are quite clear, surgery for patients with locally advanced disease remains a controversial subject that has not been fully clarified. In general, single modality therapies do not yield optimal long-term survival rates. Instead, multimodality therapy regimens are recommended in patients with an acceptable performance status, mainly consisting of a combination of chemotherapy, radiation therapy (RT), and surgery. Currently, the European Society for Medical Oncology (ESMO) recommendations for resectable locally advanced NSCLC depend on the N-status. For N2 disease that is documented intraoperatively, surgery should be followed by adjuvant chemotherapy. If single station N2 disease is demonstrated preoperatively, surgical resection with neoadjuvant or adjuvant chemotherapy is recommended. In multistation N2 or N3 disease, concurrent definitive CRT is recommended.

In more recent years, a large number of immunotherapeutic treatments and targeted therapies have been developed. An increasing number of studies have shown that incorporating these modalities in the current treatment regimens results in improved survival and treatment outcomes. As more evidence is emerging, it is likely that immunotherapy and targeted therapy will be incorporated as standard modalities in induction therapy regimens. However, despite these promising results, there is only limited data on the specific outcome in patients undergoing lung cancer surgery after immunotherapy and/or targeted therapy. Furthermore, it is not known whether there is a difference between adjuvant and neoadjuvant treatments in operable patients, if any. Although a large number of trials have focused on neoadjuvant treatment, studies such as the ADAURA and Impower010 trial have already shown the effectiveness of adjuvant treatment as well [10,11]. In this review, the latest data on surgical outcomes after induction treatment with immunotherapy and targeted therapy are discussed.

## 2. Locally Advanced Cancer

Locally advanced NSCLC represents a heterogeneous group of tumors and covers a wide spectrum of lung cancer stages. Despite efforts to define specific descriptors for subgroups of stage III NSCLC, significant heterogeneity still exists in the 8th edition of the TNM classification (Table 1) [12]. This variability is even present within the specific subgroups themselves, e.g., stage IIIA comprises patients with ipsilateral mediastinal nodal involvement, but also patients with T4 disease without nodal disease or only N1 involvement. In addition to this stage-related variability, pathological variability may produce an additional element of heterogeneity as well. Different histological subtypes often have different prognoses as biological behavior can vary substantially, especially when driver mutations are present [12,13,14,15].

Current guidelines recommend CRT for most patients with locally advanced disease. However, there is a large grey area of potentially resectable lung cancers that could benefit from treatment regimens that include surgery. The optimal therapeutic strategy and the benefits of including surgery are hotly debated topics. For these complex cases, experts recommend that personalized multimodal treatment strategies should be adopted in a multidisciplinary setting, preferentially in high-volume centers. Until recently, platinum-based chemotherapy in combination with radiation therapy (RT) was considered the gold standard in the treatment of unresectable stage III disease [16]. Typically, four cycles are administered, similar as in the adjuvant setting [3,14]. If the patient’s general condition permits, concurrent chemoradiotherapy is preferred [7]. However, despite definitive CRT, median progression-free survival (PFS) among patients who have received CRT is poor (approximately 8 months), and only approximately 15% of patients are alive at 5 years. In more recent years, consolidation immunotherapy with durvalumab has been advocated for patients with stage III, unresectable, locally advanced disease. The results of the randomized, placebo-controlled PACIFIC trial showed that durvalumab significantly increased PFS and overall survival (OS) with favorable safety profiles in this heterogeneous patient group [7,17,18].

For patients with potentially resectable locally advanced NSCLC, surgery might play a role in multimodal settings. Results from several clinical trials have shown benefits of surgery for selected patients. In the randomized-controlled Intergroup Trial 0139, definitive CRT was compared to induction CRT followed by surgical resection. A total of 429 patients were enrolled, of which 396 received treatment in one of both arms. No significant difference was found regarding OS between the two arms. However, PFS was significantly better in the surgical arm, especially in patients who had a lobectomy [19]. Similarly, the phase III ESPATUE trial attempted to evaluate the outcomes of trimodal therapy by randomizing patients with potentially resectable stage III NSCLC to either definitive CRT or induction CRT followed by surgery. In this trial, even patients with N3 disease were included. The results showed no significant difference between the surgical arm and the definitive CRT arm regarding 5-year OS and PFS. The authors concluded that both treatment strategies are acceptable for patients with resectable locally advanced disease [20]. Patients who are eligible for surgery after induction treatment with CRT, immunotherapy, or targeted therapy, have to be evaluated carefully. Technically, operations may become more challenging after induction treatments. Regarding induction CRT, previous studies have already demonstrated that neoadjuvant treatment is associated with higher rates of adverse postoperative events due to hematologic toxicity and pulmonary injury [21]. Similarly, immunotherapy or targeted therapy may give rise to more complicated resections due to pleural adhesions and fibrosis, especially in cases of major pathological response (MPR) [22,23]. However, more data is necessary to evaluate the effect of these newer modalities on operation time, conversion rates, blood loss, and perioperative complications.

## 3. Neoadjuvant Immunotherapy

### 3.1. Single Agent Immunotherapy

Forde et al. were the first to report a pilot study of 21 patients treated with neoadjuvant nivolumab in stage II-IIIA NSCLC followed by surgical resection. Toxicity was acceptable with only five adverse events (AEs), one of which was grade 3 or higher. No delays in surgery were mentioned. In a total of 20 patients out of 21, complete resection was achieved, with a MPR occurring in nine patients (45%). After 30 months, 15 out of these 20 patients were still disease-free. In two patients, presurgical radiographic chest CT-scans showed an increase in tumor size, whereas definitive pathology only showed minimal or no residual tumor cells. These findings demonstrate the possibility of obtaining clinical benefit from immunotherapy without radiographic tumor shrinkage. Furthermore, the authors investigated whether neoadjuvant PD-1 blockade could enhance the systemic priming of antitumor T-cells, thereby potentially destroying micrometastases. Earlier studies have indeed shown that PD-1 blockade enhances early T-cell activation in lymph nodes. In this study, the authors found that a large number of T-cell clones, which had expanded in the peripheral blood after PD-1 blockade, were also present in the tumor. Furthermore, patients with a MPR had significantly higher rates of these T-cell clones [24]. In addition to these findings, the authors concluded that the tumor mutational burden (TMB) was predictive of the pathological response to PD-1 blockade.

Efficacy and safety of another anti-PD-1 antibody, sintilimab, was evaluated in a phase IB trial by Li et al. Twenty-two patients with resectable stage IB-IIIA squamous NSCLC were included in this trial. MPR was achieved in 10 patients (45.5%), with a complete pathological response (pCR) rate occurring in 4 patients (18.2%). Treatment-related AEs were reported in 4 patients (18.2%), mainly grade 1 or 2. In addition, the authors reported that a decrease in standardized uptake values (SUV) on positron emission tomography/computed tomography (PET/CT) imaging prior to surgery compared to baseline, may be predictive for pathological response [25].

In the multicenter NCT02927301 trial, neoadjuvant atezolizumab is being investigated in 180 patients with resectable stage IB-IIIB NSCLC. An interim analysis of 101 patients in 2019 showed grade 3–4 AEs in 29 patients of which only 6 were treatment related. Grade 5 AEs were reported in 2 patients. However, the latter were not treatment related (cardiac death postoperatively and death due to progressive disease). Of the 90 patients undergoing surgery, 8 were excluded due to driver mutations. MPR was reported in 15 patients (18%), of which 4 (5%) with pCR. TMB was not significantly different between patients with or without MPR. Two of 26 patients (8%) with PD-L1 negative status had MPR, compared to 10 out of 35 patients (29%) with PD-L1 positive status (*p* = 0.055), indicating that PD-L1 positive levels may yield higher rates of MPR. Further safety assessment and efficacy analysis are necessary, though the preliminary data showed acceptable safety and efficacy of neoadjuvant atezolizumab [26].

Another study investigating the effects of neoadjuvant atezolizumab is the phase II PRINCEPS Trial. Preliminary results of this trial were presented during the ESMO 2020 virtual congress. In this study, neoadjuvant atezolizumab was administered to patients with stage IA-IIIA, non-N2 NSCLC. Nine patients (30%) had stage III disease. Resection was performed 20–27 days after administration of atezolizumab. In this group of 30 patients, 29 patients had a complete (R0) resection. MPR was present in 4 patients (14%). Furthermore, a correlation was observed between pathological response and PD-L1 expression at baseline. Treatment-related AEs were uncommon with only one grade 1 AE of parietal pain. The authors concluded that surgery after one dose of atezolizumab was considered safe [27].

In a feasibility study, Bott et al. examined the safety of lung resections in patients initially treated with immunotherapy for metastatic or non-resectable disease. Most common tumors were lung cancer (47%) and metastatic melanoma (37%). Most frequently used agents were nivolumab (32%), pembrolizumab (32%), and ipilimumab (16%). In total, 22 pulmonary resections were performed in 19 patients for suspected residual disease after immunotherapy. The complication rate was 32% and did not differ significantly from other reports of lung resection, such as lobectomy after induction chemotherapy or CRT. However, the authors reported dense fibrosis or adhesions to be present. R0 resection was feasible and short-term disease-free survival was acceptable. However, long term follow-up is lacking. Other limitations of the study were its retrospective nature and small simple size [22]. In a subsequent study, Bott et al. administered neoadjuvant nivolumab to 22 patients with stage I-IIIA NSCLC, of which 20 patients underwent surgery. Only one patient had an AE of grade 3 or worse. No surgical delays or deaths were reported, and a MPR was observed in 9 patients (45%). Interestingly, in more than half of the 13 attempted minimally invasive procedures, conversion was necessary due to significant hilar inflammation and fibrosis [22].

In contrast to the promising results of these previous trials, the IFCT-1601 IONESCO trial, which was presented as abstract at the ESMO 2020 congress, reported a high surgical mortality. In this trial, patients with stage IB-IIIA, non-N2 disease were given durvalumab as induction therapy. Resection was performed two to 14 days after the last administration. In total, 46 patients underwent surgery after durvalumab. Among the 46 operated patients, 9 received a pneumonectomy, 31 lobectomy, 3 bilobectomy, and 3 an exploratory thoracotomy. MPR was observed in 8 patients (18.6%), which was significantly associated with disease-free survival (DFS) at one year of follow-up. No AEs of grade 3 or higher were reported. However, 90-day mortality in this study was 9% during the interim analysis. Three patients had died postoperatively after a lobectomy; of these, one patient died of acute respiratory failure, one of a surgical complication, and the cause was unknown in one patient who died at home. One patient died after pneumonectomy due to a tracheal fistula. Due to this high 90-day mortality rate, the trial was stopped early. However, according to the authors, the high mortality is most likely related to significant comorbidities rather than durvalumab itself [28].

Currently, we await the results of the NEOMUN trial. In this study, patients with stage II-IIIA NSCLC will receive neoadjuvant pembrolizumab, regardless of mutation type. Feasibility and safety are the primary outcomes with DFS and OS as secondary outcomes [29]. Table 2 shows an overview of currently ongoing clinical trials regarding (neo)adjuvant single agent immunotherapy for resectable NSCLC.

### 3.2. Combination Therapies Including Multiple Immune Checkpoint Inhibitors

In addition to single-agent neoadjuvant immunotherapy, several studies have evaluated whether combination immunotherapy can be applied to obtain higher response rates than single-agent treatment regimens. The NEOSTAR trial compared single-agent immune checkpoint inhibitors (ICIs) to combination regimens in 44 patients with resectable stage I-IIIA (single N2) NSCLC. Patients were randomized to either nivolumab (*n* = 23) or nivolumab with ipilimumab (*n* = 21). A total of 34 patients underwent surgery, with the results demonstrating acceptable toxicity rates. One death was reported in the nivolumab arm, which was due to a bronchopleural fistula following pneumonitis. In addition, three grade 3 AEs were reported as well in the nivolumab arm. In the combination arm, only one case of grade 3 AE occurred. MPR was observed in 4 patients (17%) in the nivolumab arm and in 7 patients (33%) in the combination arm. A complete response was reported in 2 (10%) and 6 (38%) patients, respectively. Moreover, the combination therapy induced more tumor-infiltrating CD3+ and CD3+CD8+ T lymphocytes and a higher pathological response. Elevated PD-L1 levels were positively correlated with the radiographic and pathological response [30].

In a recent phase I study by Reuss et al., the authors evaluated the effects of neoadjuvant therapy with nivolumab plus ipilimumab in resectable NSCLC. The planned target of 15 patients was not reached due to toxicity, resulting in an early termination after enrolment of 9 patients. Of these 9 patients, 6 (67%) had stage III disease at the time of inclusion. Six patients (67%) had treatment-related AEs, of which half were grade 3 or higher. Three patients (33%) had tumor progression proven by histology, thereby precluding definitive surgical treatment. One patient out of 6 died postoperatively. Due to the relatively high rate of toxicity, the study was terminated early. Several explanations were suggested by the authors, such as the small sample size, long interval between neoadjuvant therapy and surgery, and a high incidence of mutations in KRAS/STK11/KEAP1 (56%), which is known to lower the efficacy of immunotherapy. A correlation was found between pathological response and PD-L1 expression. The TMB itself did not have a significant effect, in contrast to the previously mentioned report of Forde et al. in which nivolumab was used as monotherapy in the neoadjuvant setting [31]. Table 2 shows an overview of currently ongoing clinical trials regarding regimens including multiple immune checkpoint inhibitors for resectable NSCLC.

### 3.3. Combination Therapies Including Chemotherapy

In an open-label, multicenter, single-arm, phase II trial by Shu et al., the outcomes of atezolizumab plus chemotherapy in stage IB-IIIA disease were investigated in a total of 30 patients. Twenty-three patients (77%) had stage IIIA disease. Twenty-nine patients (97%) underwent surgery, of which 26 (87%) had a complete resection. No surgical complications attributable to the neoadjuvant treatment were reported. One death was reported, which was not related to the immunotherapy. The combination of atezolizumab plus carboplatin and nab-paclitaxel resulted in a MPR in up to 57% and a pCR in 33%. No significant association was reported between PD-L1 expression and MPR or pCR. However, treatment-related AEs were common, with grade 3–4 AEs occurring in 24 patients (80%). The treatment-related toxic effects were found to be manageable and did not impact on surgical resection. MPR was more prevalent in patients with squamous cell carcinoma (80%), compared to patients with adenocarcinoma (53%) [32].

In the larger, phase II, multicenter, single-arm NADIM trial, treatment outcomes of neoadjuvant chemoimmunotherapy in resectable stage IIIA NSCLC were investigated. Treatment consisted of a combination of carboplatin—paclitaxel and nivolumab. In total, 41 of the 46 eligible patients who received neoadjuvant therapy underwent surgery. Adverse events were reported in 43 patients (93%), with 14 patients having grade 3 or worse. No surgery delays or deaths were reported. R0 resection was achieved in all cases. Postoperative complications occurred in 12 patients (29%). After induction therapy, radiographic complete response was observed in 2 patients (4%) and partial response in 33 patients (72%). No progressive disease was reported. However, a MPR was observed in 34 out of 41 operated patients (83%), with pCR achieved in 26 patients (63%). Three patients (7%) had an incomplete pathological response (>10% of viable tumor cells). The two-year PFS and overall survival rates were 77.1% and 89.9%, respectively. Of the 26 patients with a pCR, 25 (96.2%) showed no progression after 2 years of follow-up. The NADIM trial also demonstrated that in tumors with baseline mutations in genes associated with poor immunotherapy prognosis, reduced PFS rates were observed [33]. The currently ongoing successor of the NADIM trial, the NADIM II trial, has been designed to investigate whether neoadjuvant nivolumab plus chemotherapy results in improved outcomes compared to chemotherapy alone. A total of 90 patients are estimated to be enrolled in this trial, with pCR as the primary endpoint. The results are expected in March 2022.

The results of the single-arm phase II SAKK 16/14 trial by Rothschild et al. were presented as well at the recent ESMO 2020 virtual congress. Pre- and postoperative durvalumab were evaluated in addition to the standard of care in stage IIIA -N2 NSCLC, irrespective of histology, mutations, or PD-L1 expression level. Sixty-eight patients were enrolled, of which 59 (87%) completed neoadjuvant immunotherapy. Radiological evaluation by PET/CT was performed after chemotherapy and immunotherapy. Complete and partial response rates on imaging increased from 4.5% to 6.5% and from 40.3% to 51.6%, respectively. Of the 55 patients undergoing surgery, 43 patients (78%) underwent a lobectomy and 5 (9%) a pneumonectomy. R0 resection was achieved in 50 patients (91%). Ten patients (18%) had a pCR and 33 patients (60%) a MPR. Postoperative nodal downstaging occurred in 37 patients (67%). One patient died postoperatively due to hemorrhage, which, according to the authors, was not related to the induction therapy. Immunotherapy in combination with chemotherapy was found to be safe in the neoadjuvant setting. The one-year event-free survival rate was 73.3% [34].

In a recently published study by Jiang et al., results on surgical feasibility after neoadjuvant immune(chemo)therapy were evaluated in 31 patients. Twenty-two patients in this study had a histological diagnosis of a squamous cell carcinoma (71%), while only 9 patients (29%) had adenocarcinoma. Several induction regimens were used depending on the specific ICI, with or without chemotherapy. Three doses were administered with a median interval of 34 days before surgery. The majority had locally advanced NSCLC (stage IIIA 52%, stage IIIB 32%), and the remainder stage II disease. Despite the heterogeneous groups necessitating cautious interpretation, the authors report low operative mortality and morbidity rates. Conversion was only necessary in 1 patient (3.2%), although only 9 patients out of 31 were operated using a minimally invasive approach. Dense adhesions around hilar lymph nodes was the main reason for conversion. Regarding the pathological response, MPR was observed in 12 patients (39%), and pCR in 3 patients (9.7%). Postoperative complications were observed in 18 patients (58%), with prolonged air leak as common complication 14 patients (45%) [35].

In addition to these (partially) published trials, several anticipated trials are still ongoing. The phase II neoCOAST trial investigates neoadjuvant durvalumab monotherapy or combined with novel agents in patients with early stage (IB-IIIA), resectable NSCLC. As in other trials mentioned previously, stage IIIA is included as well. MPR is the primary objective, feasibility of surgery 14 days after the last dose is the secondary objective [36].

The CheckMate 816 trial investigates the same population of patients with stage IB-IIIA NSCLC. Primary endpoint are pCR and event-free survival. Secondary endpoints are MPR and OS. Three arms are used; nivolumab plus ipilimumab, nivolumab plus platinum-doublet chemotherapy, and platinum-doublet chemotherapy only, all of them in a neoadjuvant setting. A total of 624 patients will be enrolled. Results are expected in May 2023 [37]. Figure 1 shows an example of a patient with a stage IIIA NSCLC treated with four cycles of neoadjuvant cisplatinum/pemetrexed and nivolumab prior to surgical resection. Final pathology revealed a pCR and the patient had an excellent recovery.

The IMpower030 is designed to assess the outcomes of neoadjuvant atezolizumab or placebo combined with chemotherapy in a planned population consisting of 374 patients with stage II, IIIA, and IIIB NSCLC disease [38].

The phase III AEGEAN trial compares chemotherapy and chemotherapy with durvalumab in a neoadjuvant setting in a planned population of 300 patients with resectable stage II and III NSCLC disease. MPR is the primary endpoint [39].

The ongoing KEYNOTE-671 phase III trial compares the gold standard platinum-based chemotherapy with pembrolizumab or placebo for stages IIB-IIIA NSCLC. In total, 786 patients are planned to be enrolled. Primary endpoints are event-free survival and OS. Secondary endpoint is MPR, pCR, safety, and patient-reported outcomes [40]. Table 2 shows an overview of currently ongoing trials of regimens including combinations of (neo)adjuvant immunotherapy and chemotherapy.

### 3.4. Combination Therapies Including Radiation Therapy

In addition to trials investigating the outcomes of combined systemic therapies, several trials have been designed to assess the possible synergistic effects of RT with immunotherapy. The NCT03237377 pilot trial is currently investigating durvalumab with or without tremelimumab added to the standard radiation scheme of 45 Gy in resectable stage IIIA NSCLC patients. RT is thought to function as an immunomodulator by upregulating antigen presentation, and lymphocyte recruitment and infiltration. In metastatic disease, it is shown to be especially effective in patients with no or low PD-L1 expression levels [2,41].

Preliminary results of the ongoing phase II NCT02904954 trial by Altorki et al. have also demonstrated an enhanced immune response by RT. Patients with stage I-IIIA NSCLC received neoadjuvant durvalumab with or without stereotactic RT. Safety was deemed acceptable, as only 4 out of 34 patients had grade 3 or 4 AEs after induction therapy, one of which was reported in the combination arm. Thirty patients received surgery, one to two weeks after the last dose of durvalumab. MPR was reported in 8 out of 17 patients (47%) in the combination group versus none in the group with durvalumab monotherapy. Moreover, when excluding four patients with epidermal growth factor receptor (EGFR) mutations, an MPR of 61.5% was achieved in the combination group [42].

In the phase II PembroX trial, the additional benefit of pembrolizumab combined with stereotactic RT in a neoadjuvant regimen is being investigated in patients with resectable stage I-IIIA NSCLC [43]. Table 2 shows an overview of currently ongoing trials of regimens including combinations of (neo)adjuvant immunotherapy and RT.

### 3.5. Neoadjuvant Targeted Therapy

Although to a lesser extent, targeted therapies consisting mainly of tyrosine kinase inhibitors (TKI) have also been evaluated in the neoadjuvant setting. The majority of data come from small case series; however, data from larger trials are emerging. Zang et al. retrospectively reported 6 cases of neoadjuvant gefitinib in locally advanced adenocarcinoma. Surgery was deemed feasible by the authors, however, no data regarding pathological response, OS or PFS was reported [44]. Liu et al. reported a case of pCR in a patient with a locally advanced, EGFR-mutated adenocarcinoma after receiving neoadjuvant gefitinib. No recurrence was reported at a follow-up of 21 months [45].

In the double-blind, phase III ADAURA trial, patients with completely resected EGFR-mutated NSCLC were randomly assigned to receive either osimertinib or placebo for 3 years. A total of 682 patients were randomized (339 in the osimertinib group and 343 in the placebo group). At 2 years, 90% of patients with stage II-IIIA disease in the osimertinib group and 44% of patients in the placebo group were alive and disease-free (hazard ratio: 0.17; 99.06% confidence interval (0.11–0.26); *p* < 0.001). The authors concluded that, for stage IB-IIIA EGFR-mutated NSCLC, DFS is significantly improved in patients that received osimertinib compared to placebo. Furthermore, no new safety concerns were noted in this study [11].

In the non-randomized CSLC-0702 trial, patients with stage IIIA-N2 were assigned to erlotinib or chemotherapy depending on the EGFR mutation status. PFS and OS did not differ, however, a higher response rate was seen in the neoadjuvant erlotinib group [46]. One promising trial is the EMERGING-CTONG 1103 trial by Zhong et al., which compares targeted therapy to conventional neoadjuvant therapy in patients with stage IIIA-N2 disease with an EGFR-mutation. A total of 71 patients were included and randomized to erlotinib (37 patients) or cisplatin plus gemcitabine (34 patients). No grade 3 or 4 AEs were encountered in the erlotinib group versus 10 patients (29.4%) in the chemotherapy group. The objective response rates were 54.1% and 34.3%, respectively. Furthermore, MPR was observed in 10% and 0%, respectively. No complete response occurred in either arm. Patients who were given erlotinib were found to have a significantly higher PFS (21.5 months versus 11.4 months in the chemotherapy group) but OS data are not fully mature yet [47].

In another, similar trial (NCT01217619), the role of neoadjuvant erlotinib was assessed in patients with stage IIIA-N2 EGFR-mutated NSCLC. A total of 19 patients were included, 14 of which underwent surgery. The radical resection rate was 68% with a 21% rate of pathological downstaging. Grade 3 or 4 AEs were encountered in 15.8% of all patients. The authors concluded that neoadjuvant therapy was well tolerated and improved the radical resection rate [48].

A recent systematic review on neoadjuvant EGFR-TKI therapy for EGFR-mutant NSCLC concluded that this provides a feasible treatment modality with satisfactory surgical outcomes and low toxicity [49]. However, further phase III trials are necessary to confirm these findings. Table 2 shows an overview of currently ongoing trials of regimens including (neo)adjuvant targeted therapy.

## 4. Discussion

In the last few years, there has been an exponential increase in studies incorporating immunotherapies and targeted therapies in neoadjuvant treatment regimens for the management of locally advanced NSCLC. A large number of trials have already reported promising results and many highly-anticipated studies are still ongoing. This substantial output of data has highlighted the heterogeneous character of NSCLC in clinical research, as many trials are including patients with both early stage and locally advanced NSCLC, regardless of histological subgroups. However, it is not clear whether grouping these different subgroups yields reliable results. For example: in a recent phase II trial of neoadjuvant atezolizumab plus chemotherapy in patients with resectable NSCLC, the results showed significant differences in MPR between squamous cell carcinoma and adenocarcinoma [32].

In general, immunotherapy seems to be well tolerated by patients. The majority of treatment-related AEs following ICI are reported to be acceptable and manageable. Reported incidence rates of grade 3 and 4 AEs are low, as evidenced by the trials published by Forde et al. (5%), Bott et al. (5%), and Wislez et al. (0%) [24,28,50]. The majority of studies report grade 3 and 4 incidence rates between 0 and 30% [51]. Future trials will hopefully elucidate which specific combinations of therapies are associated with the lowest incidence rate and severity of AEs [51]. Worth mentioning are the trials by Reuss et al. and Wislez et al., which were terminated early due to toxicity after neoadjuvant immunotherapy. In the trial of Reuss et al., combination ICI therapy resulted in AEs in 67% of patients with grade 3 or higher in 33%. Furthermore, one patient died postoperatively [31]. In the study by Wislez et al., a 90-day mortality of 9% resulted in early termination. However, according to the authors, mortality was due to comorbidities rather than durvalumab administration. The interval between the last dose of durvalumab and surgery varied from 2 to 14 days. Whether or not the mortality rate was related to the interval length is unknown. In contrast to the excessive 90-day mortality rate, no grade 3 or higher AE were seen [28].

For targeted therapies, the future seems promising as well. The recent ADAURA trial demonstrated that adjuvant osimertinib prolonged DFS in patients with resectable EGFR-mutated stages IB-IIIA NSCLC. In addition, no new safety concerns were noted in this trial. The follow-up of the ADAURA trial, the phase III, double-blind, randomized, placebo-controlled NeoADAURA trial (NCT04351555), is currently still in progress. In this trial, neoadjuvant osimertinib as monotherapy or in combination with chemotherapy is compared to chemotherapy alone in patients with resectable EGFR-mutated NSCLC [52]. Another noteworthy, ongoing trial investigating osimertinib is the phase III, double-blind, randomized, placebo-controlled LAURA trial (NCT03521154), which aims to assess the efficacy and safety of osimertinib following CRT in patients with stage III unresectable EGFR-mutated NSCLC [53]. The results of these trials are highly anticipated as they will provide more insight on the role and timing of osimertinib in this heterogenous patient group.

Other treatment regimens, such as combinations of chemotherapy with immunotherapy or targeted therapy have shown acceptable results as well. Shu et al. reported grade 3 or worse AEs in up to 75%, however, these were deemed to be manageable [32]. For combinations of RT and immunotherapy, Altorki et al. showed acceptable AEs as well, with AEs in 4 out of 34 patients, only one of which received combination immunotherapy and RT [42]. Less data is available regarding the role of targeted therapy in a neoadjuvant setting. The EMERGING-CTONG 1103 trial by Zhong et al. is the largest study investigating the role of neoadjuvant targeted therapy. Their data showed no grade 3 or higher AEs in the erlotinib group versus 29.4% in the chemotherapy group [47]. However, in a study by Xiong et al., neo-adjuvant erlotinib was associated with an incidence rate of 15.8% of grade 3 or higher AEs [48].

The potential use of immunotherapy and targeted therapy as part of neoadjuvant treatment regimens has also raised a number of concerns concerning surgical feasibility. Due to inflammatory reactions resulting in hilar fibrosis, surgical resection may be more demanding and technically challenging. Although trials reporting parameters suggestive of surgical difficulty such as operative time, blood loss, etc., are not significantly divergent at the expense of neoadjuvant ICI, higher conversion rates have been reported. Bott et al. demonstrated similar median operative times, but reported a conversion rate of 54% [50]. Jiang et al. reported a lower rate of conversion of 3.2%, however, in only 9 out of 32 patients a minimally invasive approach was attempted [35]. Preliminary results have demonstrated that surgical resection is feasible in experienced centers, but morbidity and even mortality may be significantly higher compared to classical resections without induction therapy. Caution should also be exerted in case of severe comorbidities that augment the surgical risk, especially in current smokers.

In addition to safety and feasibility, concerns about possible delays of surgery have been raised, despite the lack of data on this matter. Related to this, tumor progression with the possible risk of transitioning to an inoperable state represents another major concern. However, the majority of trials have shown favorable results regarding the pathological response and progressive disease is rarely observed.

Standardized guidelines for the assessment of response and progression are still lacking. Current radiological and pathological criteria are well known, but not validated for neoadjuvant immunotherapy, demonstrated by the issue of pseudoprogression. This was first highlighted by Forde et al., who reported two patients with radiological progression, but with almost no residual tumor cells on pathological examination [21]. This may be attributed to tumor cell infiltration. Similarly, Bott et al. reported two patients with pCR while chest CT demonstrated stable disease post-treatment [50]. The everlasting debate questioning the need for invasive restaging has therefore risen again [51]. For neoadjuvant therapies, an increasing number of studies have opted to use MPR as a surrogate endpoint of clinical efficacy. By using MPR, researchers are able to provide efficacy data within a few weeks to months after all patients are enrolled. This is in contrast to trials using OS and DFS as (sole) endpoints, which often involve years of data maturing before the read-out of results can be performed [54,55,56]. However, despite the reported validity of MPR as surrogate endpoint, the association between MPR and DFS or OS is still unclear, warranting further research [2].

Further research is necessary as many questions remain unanswered. Details regarding the optimal treatment regimens, surgical intervals, surgical morbidity, and mortality, and whether or not further adjuvant therapy is indicated, are still unclear. Furthermore, current data has still not elucidated which combination therapies are synergistic without compromising safety. Finally, the role of RT in these multimodality protocols for locally advanced disease also needs to be evaluated in future prospective trials [12,17]. It is without any doubt that immunotherapy and targeted therapies will be incorporated in our combined treatment modalities for (locally advanced) NSCLC. However, the specific role of surgery has not been determined yet and warrants further research.

## 5. Conclusions

Incorporating immunotherapy and targeted therapy as a neoadjuvant regiment in NSCLC is a hot topic in lung cancer research. Surgery for the resectable locally advanced stage after induction therapy seems feasible and safe. Although serious concerns have been raised on possible surgical difficulties due to hilar fibrosis, acceptable morbidity and mortality rates are reported in experienced centers. Immunotherapy and targeted therapies will definitely lead to a shift in management of NSCLC as numerous trials have published promising results, indicating that these therapies can improve OS and DFS without additional toxicity or reduction in quality of life. Despite this, questions regarding the optimum doses of each agent, its optimal interval from final administration to operation, and the need for adjuvant immunotherapy remain unanswered. Future studies with large sample sizes are necessary to provide more definite answers. Moreover, adequately assessing the response to ICI needs to be investigated and validated as classical radiological and pathological assessments might not be sufficient due to possible discordance. Nevertheless, it is highly likely that neoadjuvant immunotherapy and targeted therapy will create a shift in the multimodality treatment of NSCLC, especially in locally advanced stage IIIA NSCLC. Prospective phase III in the (near) future will hopefully confirm the promising preliminary results with validated outcome markers such as OS and DFS.

## Figures and Tables

**Figure 1 cancers-13-02603-f001:**
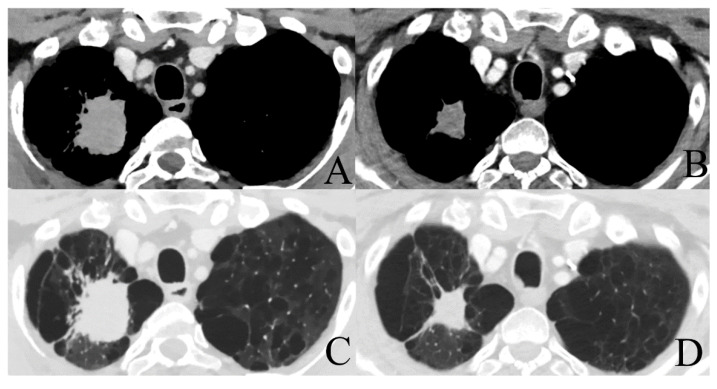
Chest CT-scan with intravenous (IV) contrast showing a stage IIIA NSCLC of 42 mm × 35 mm in the right upper lobe pre-induction therapy with a volume reduction after 4 cycles of neoadjuvant cisplatinum/pemetrexed + nivolumab (27 mm × 22 mm) ((**A**,**B**): mediastinal window; (**C**,**D**): lung window).

**Table 1 cancers-13-02603-t001:** Stage III subdivisions of non-small cell lung cancer according to Goldstraw et al. [12].

Stage	Tumor (T)	Lymph Node Status (N)	Metastasis (M)
IIIA	T1a–c, T2a–b	N2	M0
	T3	N1	M0
	T4	N0, N1	M0
IIIB	T1a–c, T2a–b	N3	M0
	T3, T4	N2	M0
IIIC	T3, T4	N3	M0

**Table 2 cancers-13-02603-t002:** Ongoing trials using (neo)adjuvant immunotherapy or targeted therapy in resectable NSCLC.

Clinical Trial	Phase	Stage	Intervention	Neoadjuvant or Adjuvant	Estimated Enrolment	Primary Endpoint
Immunotherapy monotherapy						
NCT03732664	I	IA-IIIA	Nivolumab or pembrolizumab	Neoadjuvant	40	Safety and AEs
NCT04560686	II	I-IIIB	Bintrafusp alfa	Neoadjuvant	23	MPR
NCT03197467	II	II-IIIA	Pembrolizumab	Neoadjuvant	30	AEs, radiological response, pCR
NCT02818920	II	IB-IIIA	Pembrolizumab	Neoadjuvant	35	Surgical feasibility rate
NCT04062708	II	IIIA-IIIB	Durvalumab	Neoadjuvant	55	N2 nodal clearance
NCT02994576	II	IB-IIIA (non-N2)	Atezolizumab	Neoadjuvant	60	Major toxicities or morbidities
NCT03030131 *	II	IB-IIB	Durvalumab	Neoadjuvant	81	R0 resections
NCT04379739	II	II-IIIA	Camrelizumab	Neoadjuvant	82	MPR
NCT02927301	II	IB-IIIB	Atezolizumab	Neoadjuvant + adjuvant	181	MPR
Combination immunotherapies						
NCT02259621	II	IB-IIIA	Nivolumab + ipilimumab vs. nivolumab	Neoadjuvant	30	Safety
NCT04205552	II	I-IIIA	Nivolumab vs. nivolumab + relatlimab	Neoadjuvant	60	Feasibility
NCT03794544	II	I-IIIA	Durvalumab vs. durvalumab + oleclumab or monalizumab or danvatirsen	Neoadjuvant	160	MPR
Immunotherapy + CT						
NCT04512430	II	IIIA	Atezolizumab + bevacizumab + CT	Neoadjuvant + adjuvant	26	MPR
NCT02716038	II	IB-IIIA	Atezolizumab + CT	Neoadjuvant	30	MPR
NCT03366766	II	I-IIIA	Nivolumab + CT	Neoadjuvant	34	MPR
NCT04326153	II	IIIA	Sintilimab + CT	Neoadjuvant + adjuvant	40	DFS
NCT03081689	II	IIIA (N2)	Nivolumab + CT	Neoadjuvant	46	PFS
NCT02572843	II	IIIA (N2)	Durvalumab + CT	Neoadjuvant + adjuvant	68	EFS
NCT04061590	II	I-IIIA	Pembrolizumab vs. pembrolizumab + CT	Neoadjuvant	84	Impact on tumour-infiltrating cells
NCT03158129	II	I-IIIA	Nivolumab vs. nivolumab + ipilimumab vs. nivolumab + CT vs. nivolumab + ipilimumab + CT	Neoadjuvant	88	MPR
NCT03838159	II	IIIA-IIIB	Nivolumab + CT vs. CT	Neoadjuvant	90	pCR
NCT03800134	III	II-III	Durvalumab + CT vs. CT	Neoadjuvant + adjuvant	300	MPR
NCT03456063	III	II-IIIB	Atezolizumab + CT vs. placebo + CT	Neoadjuvant + adjuvant	374	EFS
NCT04025879	III	II-IIIB	Nivolumab + CT vs. placebo + CT	Neoadjuvant + adjuvant	452	EFS
NCT02998528	III	IB-IIIA	Nivolumab + ipilimumab vs. nivolumab with CT vs. CT	Neoadjuvant	624	EFS, pCR
NCT03425643	III	II-IIIB (N2)	Pembrolizumab + CT vs. placebo + CT	Neoadjuvant + adjuvant	786	EFS, OS
NCT02595944	III	IB-IIIA	CT vs. CT + nivolumab	Adjuvant (immunotherapy)	903	DFS, OS
Immunotherapy + RT						
NCT03237377	II	IIIA	Durvalumab + RT vs. durvalumab + tremelimumab + RT	Neoadjuvant	32	Toxicity and feasibility
NCT03217071	II	I-IIIA	Pembrolizumab vs. pembrolizumab + RT	Neoadjuvant	40	Change in number infiltrating CD3+ T Cells
NCT02904954	II	I-IIIA	Durvalumab vs. durvalumab + RT	Neoadjuvant	60	DFS
NCT04245514	II	IIIA	Durvalumab + RT	Neoadjuvant + adjuvant	90	EFS
Immunotherapy + CRT						
NCT03871153	II	IIIA	Chemoradiation + durvalumab	Neoadjuvant + adjuvant	25	pCR
Targeted therapy						
NCT03433469	II	I-IIIA	Osimertinib	Neoadjuvant	27	MPR
NCT04302025	II	IIA-IIIB	CT + Alectinib or entrectinib or vemurafenib or cobimetinib or pralsetinib	Neoadjuvant + adjuvant	60	MPR
NCT04351555	III	II-IIIB	Osimertinib vs. CT vs. Osimertinib + CT	Neoadjuvant	328	MPR

* Stopped early due to excess mortality. AEs, adverse events; CRT, chemoradiation therapy; CT, chemotherapy; DFS, disease-free survival; EFS, event-free survival; MPR, major pathological response; OS, Overall survival; pCR, pathological complete response; PFS, progression-free survival; RT, radiation therapy.

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
