# Peer review of "Surgery after Induction Targeted Therapy and Immunotherapy for Lung Cancer"

_cancers, 2021, doi:10.3390/cancers13112603_

Round 1

Reviewer 1 Report

Thank you to the authors for this very complete and precise review of the literature on the  new neoadjuvant treatment of operable lung cancers and safety 

The article puts into perspective the novelties of neoadjuvant treatments by immunotherapy alone or in combination with chemotherapy, radiotherapy or dual immunotherapy or in the event of an EGFR mutation, treatment with TKi where there is less data.

I have few comments. however I think it would be useful to clarify

introduction : 

• For stages I-II (t> 4 cm) studies of chemotherapy with neoadjuvant or adjuvants have not shown a difference in overall survival, however ESMO guidelines 2017 recommend adjuvant chemotherapy (Annals of Oncology 28 (Supplement 4): iv1–iv21, 2017 doi:10.1093/annonc/mdx222)

For resectable stages 3: the type of N2 status determines the recommendations for neoadjuvant treatment (Annals of Oncology 28 (Supplement 4): iv1–iv21, 2017 doi:10.1093/annonc/mdx222)

a sentence on the uncertainties about the place and preponderance between adjuvant and neoadjuvant treatments in operable patients. note the recent advances with the ADAURA (EGFR TKI) and impower 010 (immunotherapy) studies which show the effectiveness of adjuvant treatments

page 5 line 237 specify mutations KRAS-STK11-KEAP1

paragraph3.5 : specify : we have recently made significant progress with the adaura study on the adjuvant treatment of mutated eGFR patients treated with osimertinib

discussion: few comments 

a sentence on the use of a new efficacy marker in adjuvant treatments: MPR interest, validity

conclusion :

specify the expectation of confirmation of preliminary results by prospective phase 3 studies with solid markers such as DFS, OS

Author Response

We would like to thank the reviewers for their insightful comments and suggestions. Below, we have listed a point-by-point response to the reviewer's comments. 

Comment: For stages I-II (t> 4 cm) studies of chemotherapy with neoadjuvant or adjuvants have not shown a difference in overall survival, however ESMO guidelines 2017 recommend adjuvant chemotherapy (Annals of Oncology 28 (Supplement 4): iv1–iv21, 2017 doi:10.1093/annonc/mdx222)

Answer: a sentence on ESMO guidelines regarding chemotherapy for early-stage NSCLC is added to the introduction (lines 51-53).

Comment: For resectable stages 3: the type of N2 status determines the recommendations for neoadjuvant treatment (Annals of Oncology 28 (Supplement 4): iv1–iv21, 2017 doi:10.1093/annonc/mdx222)

Answer: a few sentences on the ESMO guidelines regarding neoadjuvant treatment for resectable stage III are added to the introduction (lines 67-73)

Comment: a sentence on the uncertainties about the place and preponderance between adjuvant and neoadjuvant treatments in operable patients. note the recent advances with the ADAURA (EGFR TKI) and impower 010 (immunotherapy) studies which show the effectiveness of adjuvant treatments

Answer: a few sentences regarding adjuvant vs. neoadjuvant treatments in operable patients are added, including a mention of ADAURA and IMpower010 (lines 82-86). 

Comment: page 5 line 237 specify mutations KRAS-STK11-KEAP1

Answer: corrected (line 251)

Comment: paragraph3.5 : specify : we have recently made significant progress with the adaura study on the adjuvant treatment of mutated eGFR patients treated with osimertinib

Answer: the text has been amended to include data on the ADAURA study (lines 384-392)

Comment: a sentence on the use of a new efficacy marker in adjuvant treatments: MPR interest, validity

Answer: a few sentences on the use of MPR have been added (lines 499-506)

Comment: specify the expectation of confirmation of preliminary results by prospective phase 3 studies with solid markers such as DFS, OS

Answer: this has been specified (lines 532-533)

Reviewer 2 Report

This review covers important ongoing and completed studies in lung cancer and targeted therapy and immunotherapy from the surgeon's view. This topic of this manuscript is timely and well written.  The manuscript would be better if authors provide a nice table for the previous and ongoing studies. This is a nice review, and otherwise I do not have further comments.

Author Response

We would like to thank the reviewers for their insightful comments and suggestions. Below, we have listed a point-by-point response to the comments of the reviewers.

Comment: ...The manuscript would be better if authors provide a nice table for the previous and ongoing studies. This is a nice review, and otherwise I do not have further comments.

Answer: thank you for this suggestion. We have added a table including all ongoing (neo)adjuvant immunotherapy and targeted therapy trials (Table 2). Due to the large number of available trials, we have opted to include only currently ongoing trials in this table. The trials that have been published already are discussed in the text.